# Information sharing in high-dimensional gene expression data for improved parameter estimation in concentration-response modelling

**Franziska Kappenberg**◉*, **Jörg Rahnenführer**

Department of Statistics, TU Dortmund University, Dortmund, Germany

* kappenberg@statistik.tu-dortmund.de

**Data Availability Statement:** The original data was published by Krug et al (2013, https://doi.org/10.1007/s00204-012-0967-3). All relevant data for

## Abstract

In toxicological concentration-response studies, a frequent goal is the determination of an 'alert concentration', i.e. the lowest concentration where a notable change in the response in comparison to the control is observed. In high-throughput gene expression experiments, e.g. based on microarray or RNA-seq technology, concentration-response profiles can be measured for thousands of genes simultaneously. One approach for determining the alert concentration is given by fitting a parametric model to the data which allows interpolation between the tested concentrations. It is well known that the quality of a model fit improves with the number of measured data points. However, adding new replicates for existing concentrations or even several replicates for new concentrations is time-consuming and expensive. Here, we propose an empirical Bayes approach to information sharing across genes, where in essence a weighted mean of the individual estimate for one specific parameter of a fitted model and the mean of all estimates of the entire set of genes is calculated as a result. Results of a controlled plasmode simulation study show that for many genes a notable improvement in terms of the mean squared error (MSE) between estimate and true underlying value of the parameter can be observed. However, for some genes, the MSE increases, and this cannot be prevented by using a more sophisticated prior distribution in the Bayesian approach.

## Introduction

In toxicological research, concentration-response analyses are an integral part of understanding the properties of compounds. Typical endpoints include the viability of cells treated with increasing concentrations of the specific compounds, protein measurements, or gene expression measurements. Often, the goal is the determination of an 'alert concentration', i.e. the lowest concentration where a notable change in response in comparison to the control is observed. This alert concentration can be determined based on multiple testing against the control, or based on the fitting of a parametric curve. Examples for observation-based alert concentrations, i.e. where multiple comparisons of the actually considered concentrations

reproducing the results are uploaded to the GitHub repository at https://github.com/FKappenberg/Paper-InformationSharingAcrossGenes.

**Funding:** FK, JR were supported (in part) by the Research Training Group "Biostatistical Methods for High-Dimensional Data in Toxicology" (RTG 2624, P1) funded by the Deutsche Forschungsgemeinschaft (DFG, German Research Foundation - Project Number 427806116). The funders had no role in study design, data collection and analysis, decision to publish, or preparation of the manuscript.

**Competing interests:** The authors have declared that no competing interests exist.

against the negative control are performed, are given, among others, by the LOEC (lowest observed effect concentration) and the NOEC (no observed effect concentration) [1].

When moving to parametric modelling, calculating alert concentrations also outside of the measured concentrations is possible. One commonly used approach is given by the effective concentration (EC value), which corresponds to the concentration where a pre-specified percentage of the maximal observed effect is attained [2]. For cell viability experiments, where the measured response is given by some percentage, EC values are typically calculated in an absolute way, i.e. as the concentration, where the fitted curve attains the specific pre-defined percentage. In applications such as gene expression data, where the responses themselves do not correspond to percentages, EC values are calculated in a relative way. In these cases, a certain percentage of the overall effect, i.e. the difference in the response values between the highest and lowest measured concentration (or between the two asymptotes of a fitted model), needs to be attained by the fitted curve. In some applications, analogously to the concept of the LOEC, the lowest concentration is of interest where some effect in comparison to the control is observed. Such concentrations can be found via parametric modelling and the subsequent calculation of an alert concentration such as the LEC (lowest effective concentration, [3, 4]) or via the BMD approach (benchmark doses, [5]).

Especially for gene expression as considered endpoint, due to high-throughput technology such as microarray technology, RNA-seq or TempO-seq data [6], often many genes are considered simultaneously for only a very small number of samples. Adding additional replicates for the already chosen concentrations or even several replicates for new concentrations is very time-consuming and expensive. However, the quality of fitted curves improves with more available data.

In this work, we propose a new method, using information sharing across genes in order to improve the estimation of an alert concentration. This approach is a relaxation of the approach of common parameters. [7], e.g., have shown some improvement in statistical inference for dose-finding studies when parameters are shared across treatments. However, they restrict the information sharing to location and scale parameter and even reason that the assumption of equality for the here considered parameters might be too strong.

The alert concentration of interest considered here is the $EC50$, i.e. the concentration where half of the maximal observed effect is attained. In the situation of gene expression as response data, as considered in this work, the $EC50$ is to be understood in a relative way. In the chosen four-parametric log-logistic model (see, e.g., [2]) used for the fitting of a parametric curve, this value is directly included as a parameter. Especially for smaller data sets, a parameterization of the $EC50$ value on log-scale is proposed to better meet the underlying assumption of normality for the parameters. One of the here proposed approaches works via approximating the underlying distribution of the EC50 values for one data set with a mixture of normal distributions. Such an approach has been used before in [8], in a simpler way, where a mixture of two normal distributions was fitted for $ED90$ (here referring to doses instead of concentrations) values on log-scale, where the response value was given by the biomass of plants after treatment with different doses of a herbicide.

In order to share the information about the parameter estimate of the log $EC50$ across genes, an empirical Bayes approach, based on the normal-normal model, is used. In brief, the log $EC50$ is assumed to follow a normal distribution with mean $\mu$, and this parameter $\mu$ is also assumed to follow a normal distribution. The posterior, given an observed value of the log $EC50$ for one gene, is then again given by a normal distribution, where the mean is a weighted mean of the observed value and the mean of the prior distribution. The parameters of the prior distribution for $\mu$ are estimated from the data itself, which explains the name 'empirical Bayes' [9].

The empirical Bayes framework is well-established in the context of high-dimensional gene data analysis, e.g. using the common limma (linear models for microarray data, [10]) approach. In this approach, a moderated *t*-test is performed, where the individual variance estimate of a gene is adjusted using combined information of all genes considered. In contrast to the approach proposed here, which aims at improving the estimation of an alert concentration based on a parametric model, the limma approach is used for the calculation of differentially expressed genes between two test conditions. As another example, [11] propose an empirical Bayes approach for differentially expressed genes tailored to time-course data based on microarrays.

Our approach is based on the frequentist approach to fitting the underlying dose-response model. However, Bayesian approaches for Bayesian fitting (even in an hierarchical way) of dose-response models exist, as presented e.g. in [12–14].

This paper is structured as follows: First, the underlying parametric model for the nonlinear curve-fitting and the numerical procedure to perform this fitting are introduced. Then, the empirical Bayes approach for sharing information across genes is proposed, in three versions, based on three different assumptions for estimating the prior distribution. In a controlled simulation study, these three methods were compared to a baseline method without shared information between genes, i.e. where only individual curve-fitting per gene is performed. As target variable, the parameter from the assumed concentration-response relationship denoting the *EC*50 was used. The quality of the methods was assessed in terms of the mean squared error between the estimated parameter value and the known real, underlying parameter value. Finally, the Bayesian approaches were applied to the real data case study [15], which was also used as basis for the simulation study.

## Materials and methods

### Statistical methods

A variety of models exists for describing the relationship between the concentration $x$ and a response values $y$, e.g. the family of log-logistic models, the family of log-normal models, and Weibull models [16, 17]. When assuming a sigmoidal form of the relationship, a popular model is the *four parameter log-logistic* (4pLL) model. For the concentration $x$ with $x \geq 0$ and a parameter vector $\phi := (b, c, d, e)$ with $e \geq 0$, this model is defined as

$$f(x, \phi) = c + \frac{d - c}{1 + \exp(b(\log(x) - \log(e)))}. \tag{1}$$

Often, especially for small data sets, the re-parameterization $\tilde{e} := \log(e)$ is used [2]. The parameters $c$ and $d$ correspond to the lower and upper asymptote of the curve, $b$ is a parameter proportional to the slope of the curve, and $e$ is the concentration at which the half-maximal effect is attained. This concentration is also called the *EC*50, the effective concentration where 50% of the maximal observed effect is observed. Since this is a meaningful concentration where a relevant change in gene expression can be observed, in the following, this parameter in its logarithmic parameterization will in the following be the target estimate.

For $x_1, \ldots, x_p$ the concentration values (equal concentrations are allowed) and $y_1, \ldots, y_p$ the corresponding observed response values, it is assumed that $y_i$ is the observation of a normally distributed random variable $Y_i$ with mean $f(x_i, \phi)$ and fixed variance $\sigma^2$ for $i = 1, \ldots, p$. The parameters $\phi$ are estimated via minimizing the following sum of squared errors:

$$\sum_{i=1}^{p} (y_i - f(x_i, \phi))^2.$$

This is achieved using a numerical Quasi-Newton method. $(1 - \alpha)$-confidence intervals for the parameters are obtained in the typical way by calculating

$$\hat{\phi}_i \pm K \cdot \hat{\text{se}}(\hat{\phi}_i),$$

where $\phi_i$ is one of the parameters $b, c, d, e, \tilde{e}$, and $K$ is the $1 - \alpha/2$-quantile of a $t$-distribution with $p - 4$ degrees of freedom [2].

The alert concentration of interest is the log-transformed $EC50$, i.e. the parameter $\tilde{e}$ in the parameterization of the 4pLL model from Eq (1). Only this parameter from the 4pLL model is thus considered for the Bayesian information sharing. The random variable $X$ corresponding to the parameter $\tilde{e}$ is assumed to follow a normal distribution for the Bayesian information-sharing approach:

$$X|\mu \sim \mathcal{N}(\mu, \sigma^2),$$

where a normal distribution is assumed as prior distribution for parameter $\mu$, specifically

$$\mu \sim \mathcal{N}(\mu_0, \tau^2).$$

It follows that the posterior distribution for $\mu|x$, where $x$ is the observed value $\hat{\tilde{e}}$ for one specific gene, is a normal distribution with

$$\mu|x \sim \mathcal{N}\left(\frac{\tau^2 x + \sigma^2 \mu_0}{\tau^2 + \sigma^2}, \frac{\tau^2 + \sigma^2}{\tau^2 + \sigma^2}\right). \tag{2}$$

Thus, the resulting mean of the posterior distribution is a weighted mean of the original observation $x$ and the prior mean $\mu_0$. $(1 - \alpha)$-credible intervals are obtained via the $\alpha/2$ and the $1 - \alpha/2$ quantiles of the posterior distribution.

An *empirical Bayes approach* is chosen, where the prior parameters $\mu_0$ and $\tau^2$ are also estimated from the data. A total number of $n$ genes is assumed to be considered simultaneously, yielding individual estimates $\hat{\tilde{e}}_1, \ldots, \hat{\tilde{e}}_n$, which then can be used for estimating the prior parameters. Specifically, two approaches for estimating the prior parameters are considered here: In the maximum-likelihood approach (ML-approach), $\mu_0$ and $\tau^2$ are estimated via the empirical mean and the empirical variance of all estimates $\hat{\tilde{e}}_1, \ldots, \hat{\tilde{e}}_n$, respectively. The second approach considers a robust estimation of the prior parameters (robust approach): $\mu_0$ is estimated as the median of $\hat{\tilde{e}}_1, \ldots, \hat{\tilde{e}}_n$. For the robust estimation of $\tau^2$, the median absolute deviation (MAD) of $\hat{\tilde{e}}_1, \ldots, \hat{\tilde{e}}_n$ is calculated and multiplied with the factor 1.4826 to ensure consistency for the here assumed normal distribution. The result of this multiplication is then squared. The parameter $\sigma^2$ is individually calculated as the squared standard error of the estimates $\hat{\tilde{e}}$ for all genes.

Additionally, a more complex but also more flexible prior is considered: It is assumed that the empirical prior follows a mixing distribution of five normal distributions. For the estimation of the prior parameters, therefore the estimation of a mixing model of the form

$$f(y_j; \Psi) = \sum_{i=1}^{5} \lambda_i \cdot f_i(y_j; \theta_i)$$

with $\sum_{i=1}^{5} \lambda_i = 1$ is required. Here, $y_j, j = 1 \ldots, n$ denote the observed values and $\Psi = (\theta_1, \ldots, \theta_5, \lambda_1, \ldots, \lambda_5)^\top$ the parameter vector. Each $f_i$ denotes the density function of a normal distribution with parameter vector $\theta_i = (\mu_i^{(0)}, \tau_i^2)$. It is necessary to both estimate the parameters $\theta_i$ of the individual distributions, as well as the mixing parameters $\lambda_i$. This is achieved by employing

**Table 1. Overview of the three approaches to perform the Bayesian information-sharing.**

| Approach | Prior distribution | Estimation of prior mean | Estimation of prior variance |
|---|---|---|---|
| ML | $\mu \sim \mathcal{N}(\mu_0, \tau^2)$ | Empirical mean | Empirical variance |
| Robust | $\mu \sim \mathcal{N}(\mu_0, \tau^2)$ | Median | MAD |
| Mixing distribution | Mixture of 5 normal distributions | EM-algorithm | EM-algorithm |

the *expectation maximization algorithm* (EM algorithm), which alternates between the assignment of the observations to the classes, which here are distributions, and the estimation of the parameters of the distribution [18], see [19] for a detailed description of the algorithm.

With a mixture of 5 normal distributions as prior distribution for $X|\mu$, in the normal-normal model, the posterior is again a mixture of 5 normal distributions. It has a closed form, which is a mixture of normal distributions as in Eq 2, where additionally posterior values for the mixing parameters need to be calculated.

The three proposed approaches, with respective assumed prior distributions and estimation of the prior parameter values, are summarized in Table 1.

## Real data case study

The real data example, which is also the basis for the plasmode simulation study, is a case study that was conducted to investigate the development of human embryonic stem cells (hESC) to neuroectoderm [15]. Cells were treated *in vitro* with valproic acid (VPA) at seven different concentrations (25, 150, 350, 450, 550, 800, and 1000 µM), where each concentration was assessed in three replicate experiments. Additionally, six replicates for the negative control (untreated) were measured.

The study was carried out within the ESNATS (Embryonic Stem cell-based Novel Alternative Testing Strategies) project, which was funded by the European Commission. ESNATS targeted the prediction of toxicity of drug candidates. Gene expression data was obtained with Affymetrix Microarray technology, using the GeneChip R Human Genome U133 Plus 2.0 [20]. This resulted in measurements of 54675 probe sets for each experiment. Preprocessing was performed with the robust multi array analysis (RMA) algorithm [21], which includes the three steps background correction, normalisation and summarising the data to one value. The same parameters as in the original case study [15] are used for preprocessing.

## Simulation study

The empirical Bayes method for information sharing across genes was assessed in a controlled simulation study. In order to include real biological correlation structures to the simulated datasets, a so-called *plasmode simulation study* was conducted. The basic idea is, that in addition to retaining the true structure of an underlying dataset, the data is manipulated in a way such that true effects are known [22].

The simulation study was based on the VPA dataset from the real data case study. From all 54675 probe sets measured, those fulfilling the following two conditions were selected:

1. Statistical significance: When performing a one-way analysis of variance (ANOVA) for each probe set separately, only those are considered further where the unadjusted $p$-value is smaller than 0.001.

2. Biological relevance: The range covered by the expression values needs to be at least $\log_2(1.5) \approx 0.585$, and the direction of the profile needs to be unambiguous. The first

constraint means that for at least one concentration, the absolute value of the difference in mean between the expression value for this concentration and the expression value for the control (i.e. the $\log_2$-fold change) needs to exceed $\log_2(1.5)$. The second constraint means that not simultaneously for one concentration the $\log_2$-fold change is larger than $\log_2(1.5)$ and for another it is smaller than $-\log_2(1.5)$.

Selecting probe sets according to these criteria yields 7191 probe sets as candidates. These 7191 probe sets measured with Affymetrix technology represent genes. In the following, to avoid mixing the terms probe set and gene, and since the general concept can be also be applied to other types of gene expression measurements, we use the term gene also for probe sets.

A 4pLL model was fitted to each gene, resulting in a vector of the four parameters $b, c, d, \tilde{e}$ for each gene. These parameters were then used as true, underlying parameters. The following procedure was repeated 1000 times: The individual true 4pLL models, based on the true underlying parameters, were evaluated at the concentrations 0, 25, 150, 350, 450, 550, 800, and 1000, according to the concentrations of the real VPA dataset. Normally distributed noise with mean 0 and standard deviation 0.1 was added in six replicates to the control and in three replicates to all non-control concentrations, yielding a simulated expression dataset with 27 observations for each gene.

For each of these simulated datasets, again a 4pLL model was fitted. The corresponding fitted parameter $\hat{\tilde{e}}$ was considered as the *direct estimate* of parameter $\tilde{e}$ for each gene, respectively. For each simulated gene, the three approaches of the Bayes procedure (ML estimation, robust estimation, and mixing estimation, see Table 1) were applied, yielding three posterior distributions for each gene. The means of the respective posterior distributions were then considered as the Bayesian estimates of parameter $\tilde{e}$, corresponding to the three approaches.

## Software

All analyses were performed in the statistical programming language R, version 4.1.2 [23]. For fitting dose-response models, the package `drc`, version 3.0–1, [24] was used. The Bayesian analyses were conducted using the package `LearnBayes`, version 2.15.1, [25], and mixing distributions were estimated using the package `mixtools`, version 2.0.0, [26]. For graphical display, the package `ggplot2`, version 3.4.0, [27] was used.

The R code and the data needed for reproducing the simulation study are available via the Github repository https://github.com/FKappenberg/Paper-InformationSharingAcrossGenes.

## Results

### Descriptive analysis of the VPA dataset

First, 4pLL models were fitted to the 7191 original genes. Histograms of the resulting parameter estimates are shown in S1 Fig. Estimates for the parameter $\tilde{e}$, together with a normal distribution fitted to these values, are of particular interest with respect to the following analyses of the Bayes method. The estimated normal distributions, once based on the maximum-likelihood (ML) estimation via empirical mean and empirical variance (approach 1) and once based on the robust estimation via empirical median and empirical MAD (approach 2), together with a histogram of the parameter values of $\tilde{e}$, are shown in Fig 1(A). The ML estimation yields a far larger variance of the density function, with relatively heavy tails, while the robust estimation is more narrow and thus has higher density values in the middle range of the curve. However, both curves do not fit the data particularly well.

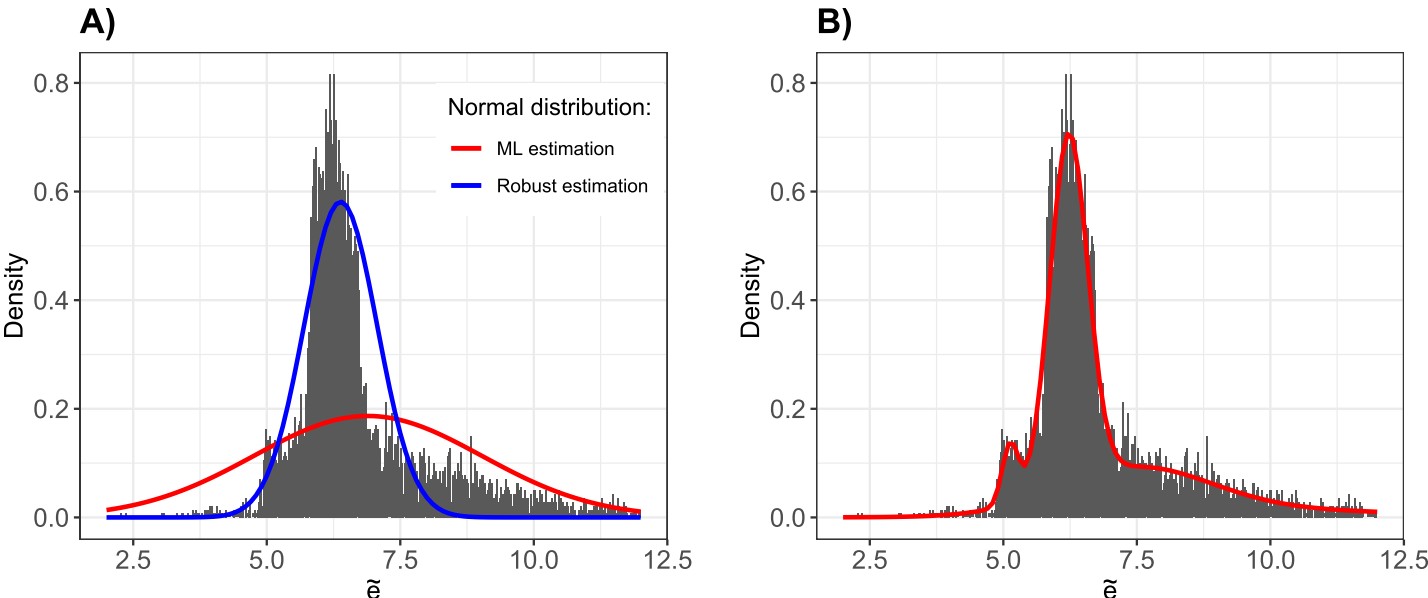

**Fig 1. Fitted normal distributions.** Different distributions are fitted to the set of estimates for parameter $\tilde{e}$ for the 7191 genes. (A) A univariate normal distribution is fitted, once estimating mean and variance (red curve), and once estimating the robust counterparts median and MAD (blue curve). (B) A mixture of 5 normal distributions is fitted with the EM algorithm, the curve shows the resulting (mixture) density function.

Via the EM algorithm, a mixture of 5 normal distributions (approach 3) was fitted to the values of parameter $\tilde{e}$ for the 7191 selected genes. Equal initial values for the mixing proportions were used, with starting values for the means given by the vector (6.2, 5, 8, 12, 8) and for the standard deviations by the corresponding vector (0.3, 0.5, 0.5, 1, 0.3). Using exactly five normal distributions is motivated as follows. One distribution is used for modelling the middle range of the distribution of $\tilde{e}$, two distributions are responsible for the heavy tails, respectively, and the remaining two distributions can represent any artefacts in high and low values that may be observed.

The resulting parameter values are summarized in Table 2. A visual display of the resulting density function is given in Fig 1(B), where an overall good fit of the mixture normal distribution to the histogram of $\tilde{e}$ can be observed, a clear improvement compared to the other two approaches. The individual density functions are displayed in S2 Fig.

### Results of the simulation study

The simulation study was conducted as described above. Due to numerical problems, sometimes no 4pLL model could be fitted to a simulated expression data set, or missing values were obtained in the Bayes method for the mixture normal distributions due to non-convergence of

**Table 2. Resulting values of the EM algorithm.**

| Component of mixing model | 1 | 2 | 3 | 4 | 5 |
|---|---|---|---|---|---|
| Mixture weight | 0.570 | 0.042 | 0.093 | 0.002 | 0.294 |
| Mean | 6.225 | 5.128 | 9.755 | 19.092 | 7.458 |
| Standard deviation | 0.352 | 0.150 | 2.643 | 32.145 | 1.399 |

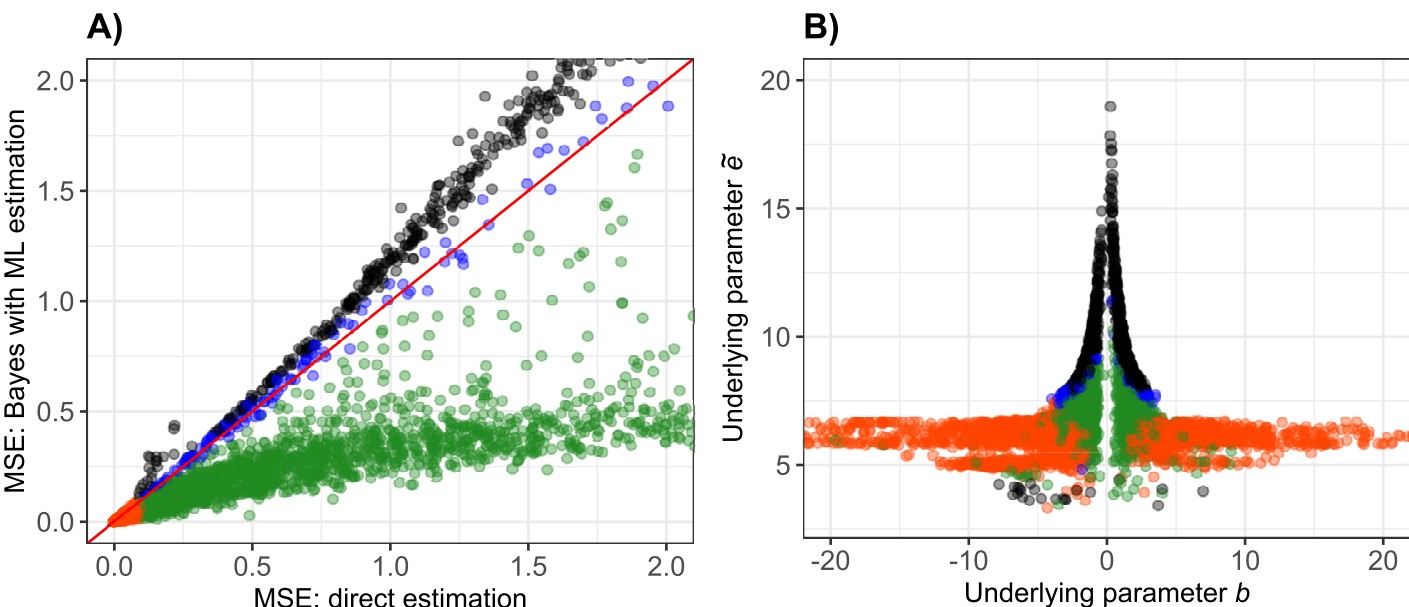

**Fig 2. Comparison of MSE for the direct estimation and for the Bayes approach based on ML estimation (A), together with true underlying parameters $b$ and $\tilde{e}$ (B).** Points represent genes, and they are colored according to the comparison of the performance of the two approaches (orange: MSEs very small, blue: MSEs comparable, green: MSE smaller for Bayes approach, black: MSE smaller for direct approach). The underlying parameter values are colored in the same way.

the EM algorithm. For the analyses, only those genes were considered for which missing values occurred in at most 200 out of the 1000 simulation runs, leaving 6891 genes in the analysis.

In order to compare the results from the Bayes approaches to the direct estimation of parameter $\tilde{e}$, mean squared errors (MSE) across the simulation runs were calculated. For this, the respective direct or Bayesian estimates were compared to the true underlying parameter $\tilde{e}$ used for the simulation. For each gene, only those simulation runs were considered, in which for the respective compared method an estimate was obtained.

The resulting MSEs for the comparison of the direct estimation and the Bayes approach based on ML estimation are shown in Fig 2(A). Each point represents one gene, and the red diagonal line represents the case where the MSEs for both methods are equal. Points are colored in orange, if both MSEs are smaller than 0.1, i.e. the MSEs are negligibly small. Points are colored in green, if the MSE based on the new Bayes approach is smaller than the MSE based on the direct approach by a factor of at least 1.1, and colored in black in the opposite case. The remaining points, i.e. when none of the approaches performs notably better than the other, are colored in blue.

To understand which factors influence the result of the MSE comparison, in Fig 2(B) the underlying, shape-defining parameters $b$ and $\tilde{e}$ of the 4pLL model used for the simulation are colored according to the results of the MSE comparison. The Bayes method performs worse than the direct method for gene with a comparatively large value of parameter $\tilde{e}$ (black points), and better for genes with a value of parameter $b$ close to zero, i.e. for genes with a rather flat slope (green points).

Corresponding results for the robust estimation of the prior distribution, and for the flexible estimation of the prior distribution with a mixture model are shown in S3 and S4 Figs.

The effect of the three Bayes approaches in comparison to the direct approach is quantified by the number of genes with low, better, similar, or worse MSE results, see Table 3. The ML

**Table 3. Effect of the three Bayes approaches on the MSE of individual genes, in comparison to the direct estimation of parameter $\tilde{e}$.**

| Approach | Low MSE (orange) | Better (green) | Similar (blue) | Worse (black) |
|---|---|---|---|---|
| 1. ML | 3402 | 2281 | 166 | 1042 |
| 2. Robust | 3371 | 2045 | 52 | 1423 |
| 3. Mixing distribution | 3375 | 2113 | 50 | 1353 |

approach yields a slightly larger number of improvements, compared to the robust and the mixing distribution approach, whose results are very similar to each other.

The parameters for the corresponding prior distributions in all 1000 simulation runs are shown in S5 Fig for the ML and the robust approach and in S6 Fig for the mixing distribution approach. Briefly, as observed for the original data set, the ML estimates are larger than the corresponding robust estimates, both for the mean value and for the standard deviation.

To assess the strength of the effect of the Bayes procedures, MA-type plots for all three different versions are shown in Fig 3 ([28], adapted from [29]). In these plots, on the x-axis, the product of the resulting MSEs for the direct estimation and the respective Bayes approach is displayed, and on the y-axis, the ratio of these MSEs is plotted. Both the product and the ratio are shown on log-scale. Colors are obtained from the comparison of the MSEs, i.e. as in Table 3. An MSE improved by the Bayes approach corresponds to a negative ratio, and more extreme values of the ratio correspond to comparatively far better results.

Above it was reported (Table 3) that the ML-approach leads to a larger number of genes with improved MSE. However, the plots in Fig 3 demonstrate that for many genes the robust and the mixing distribution approach lead to a more extreme improvement. Especially for the robust Bayes approach, the ratio of the MSEs becomes very small for some genes, indicating that in terms of the MSE, the Bayes approach yields a very strong improvement.

In the data set analyzed here, the maximum concentration value, for which gene expression values were measured, is 1000. Since $\log(1000) = 6.91$, all values of $\tilde{e}$ that are larger than 6.91

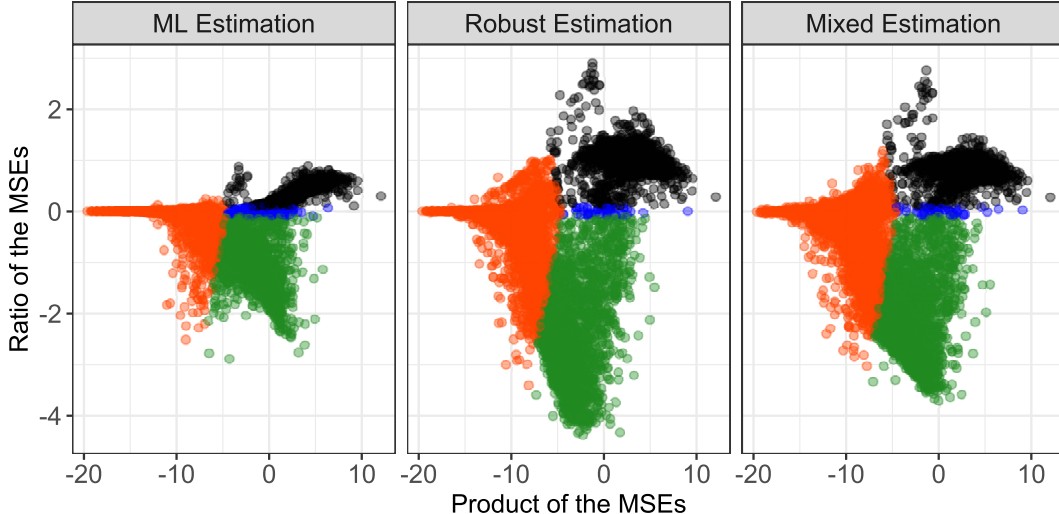

**Fig 3. MA-type plots showing the results of the three Bayes approaches in comparison to the direct estimation.** On the x-axis, the product of the resulting MSEs for the direct estimation and the respective Bayes approach is displayed, and on the y-axis, the ratio of these MSEs is plotted, both on log-scale. Colors are obtained from the comparison of the MSEs, i.e. as in Table 3.

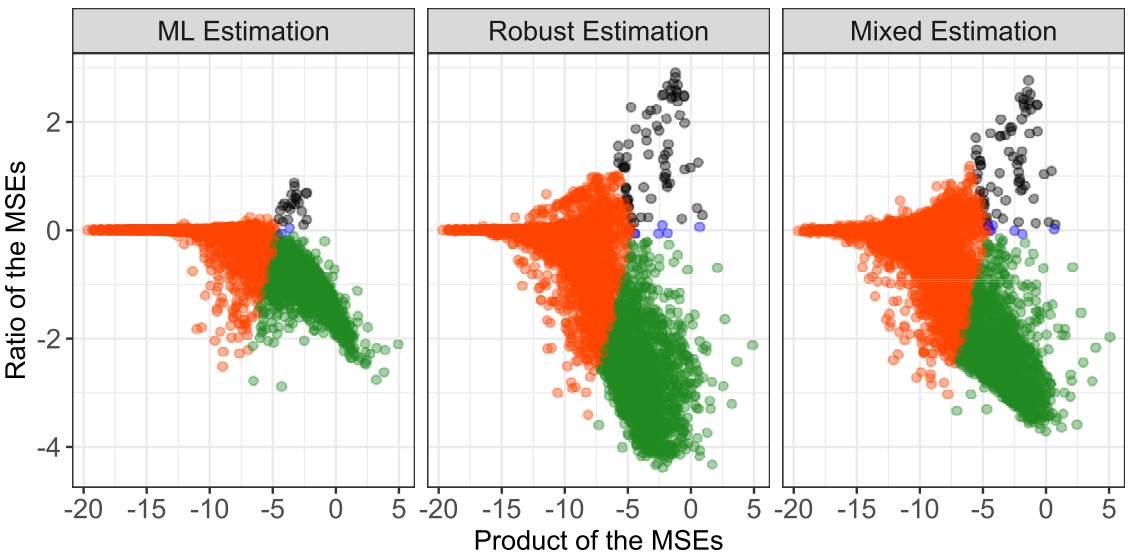

**Fig 4. MA-type plots showing the results of the three Bayes approaches in comparison to the direct estimation.** In comparison to Fig 3, here only genes with true underlying value of parameter $\tilde{e}$ smaller than 6.91 are considered.

correspond to curves where the inflection point is estimated at a larger concentration than the maximum tested concentration. This indicates an overall unfeasible and unreasonable curve fit, thus, these cases should be interpreted with caution anyways.

In Fig 4, the same MA-type plots as before are shown, but now restricted to those genes where the true underlying value of $\tilde{e}$ is smaller than 6.91. The Bayes procedure, however, is still based on the entire set of genes.

Both from the plots of the distribution of the true underlying parameter $\tilde{e}$ (Fig 2, S3 and S4 Figs) and from the restricted MA-type plots it can be seen, that not considering genes with an unreasonably high value of $\tilde{e}$ leads to far fewer black and blue dots in the plot. This means that the number of genes for which the estimation of $\tilde{e}$ is deteriorated is clearly reduced. However, some genes that previously showed an improvement in the Bayes method are now also no longer considered, but this applies mostly to genes with only small to moderate improvement, i.e. with a value of the ratio close to 0.

Next, briefly, coverage probabilities (CP) of the credibility intervals for parameter $\tilde{e}$ are compared, between the direct estimation and the Bayes approach with ML estimation. Fig 5 (A) shows a histogram of the CP for the direct estimation and a comparison between these and the ones for the Bayes approach with ML estimation. A confidence level of 0.95 was used for calculating confidence intervals, but it turns out that for the direct estimation the CPs are as low as 0.6. Fig 5(B) shows a scatterplot for the comparison of the CPs with those obtained with the Bayes ML approach. Only for a small subset of genes (colored black), the CPs for the Bayes ML approach, the CPs are considerably lower. For those genes, MSE was higher for the Bayes ML approach. However, very similar CP values can be observed for those genes for which the MSE was very small in both approaches (red) or even smaller in the Bayes approach (green). Thus, an improvement of MSE does not come at a cost of lower CP.

## Application

The four methods for parameter estimation in 4pLL models (direct, ML (Bayes), robust (Bayes) and mixing distribution (Bayes)) were applied directly to the data from the real case

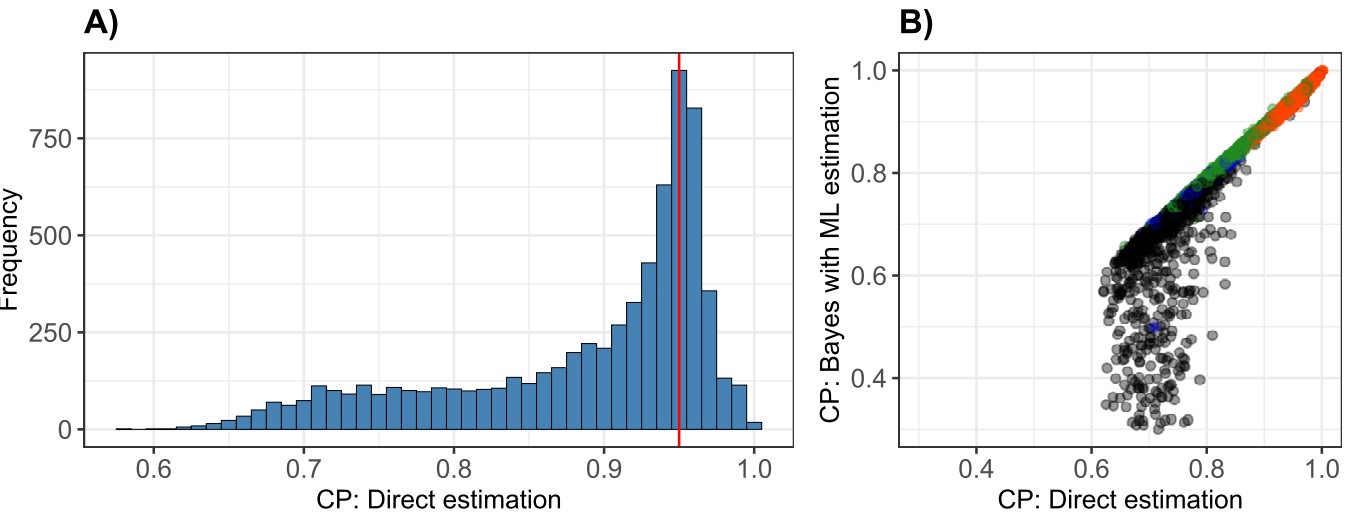

**Fig 5. Coverage probabilities (CP), based on confidence intervals for direct estimation and on credibility intervals for ML Bayes estimation.** (A) Histogram of CPs for direct estimation, the vertical red line indicates the confidence level 0.95. (B) Comparison of CPs for direct and ML Bayes estimation. Colors of points are the same as in Fig 2 (orange: MSEs very small, blue: MSEs comparable, green: MSE smaller for Bayes approach, black: MSE smaller for direct approach).

study to compare the resulting estimates. Fig 6 displays scatterplots of the estimates for parameter $\tilde{e}$, comparing the three Bayes approaches against the direct approach, respectively. The blue horizontal line indicates the mean of the prior normal distribution, estimated via the mean (ML estimate, value 6.895) or the median (robust estimate, value 6.383) of all direct estimates, respectively. Since the mixed prior is based on five normal distributions, indicating one overall mean would not be meaningful in this case.

The shrinkage of the direct estimates towards the prior mean values is clearly visible for all three approaches. Shrinkage is overall stronger for the robust and the mixed estimation than for the ML estimation. Larger values tend to be shrunken more than smaller values, indicating a generally larger uncertainty in the estimation of parameter $\tilde{e}$ when this value is large.

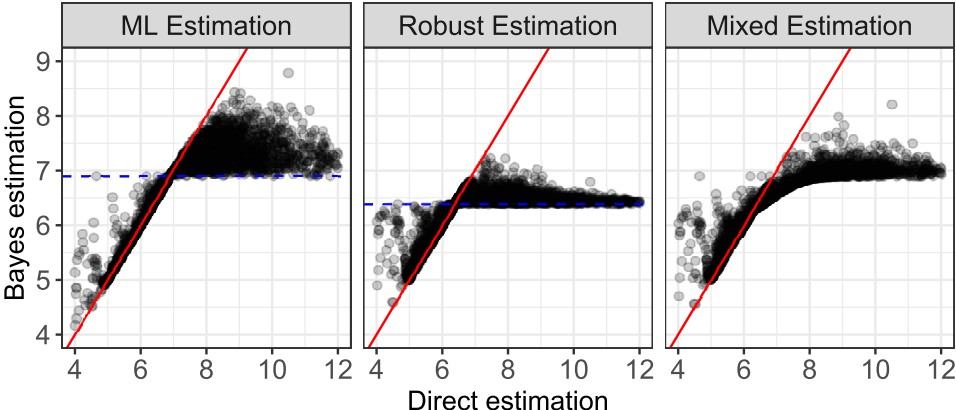

**Fig 6. Comparison of the estimates for parameter $\tilde{e}$ for the direct estimation against all three Bayesian approaches.** The dashed blue lines indicate the mean and median, respectively, used for the specification of the prior normal distribution.

## Discussion and conclusion

The calculation of an alert concentration as a part of general concentration-response analyses is an important aspect in toxicological research. Observation-based alert concentrations include concepts as the LOEC (lowest observed effec concentration) or the NOEC (no observed effect concentration) [1] or alert concentrations such as ED-values [2] or benchmark doses [5]. Especially when considering high-throughput gene expression experiments, many (often thousands of) dose-response curves are considered simultaneously, and some similarity between the resulting alert concentrations is biologically plausible.

Thus, in this paper, a method is proposed to share information across genes in order to improve the estimation of the $EC50$, i.e. the concentration where half of the maximal effect is obtained, as alert concentration. The method is based on an empirical Bayes approach, where the estimate for the logarithmic $EC50$ is assumed to follow a normal distribution with mean $\mu$ which is assumed to follow a normal distribution as well. Parameters of this prior distribution are directly estimated from the data, either using the empirical mean and empirical variance, the median and the MAD, or an approach via a mixture of 5 normal distributions. For these modelling approaches, the posterior given an observed value for the logarithmic $EC50$ again follows a normal distribution with a mean value that is essentially a weighted mean of the prior mean and the observed value.

Results of a controlled plasmode simulation study, based on data from a real data case study [15], showed that the estimate of the log $EC50$ is improved in terms of MSE for a notable number of genes, for each of the three approaches to calculate the prior. The maximum likelihood prior leads to the largest number of improvements, while the individual improvements are generally larger for the other two approaches. This does not come at a cost of lower coverage probabilities for the corresponding confidence and credible intervals. When excluding genes from the analysis for which an initial fit with the chosen log-logistic model does not lead to a plausible result, the ratio of genes with clear improvement is further increased.

In this work, only the functional form of a log-logistic model to describe the relationship between concentration and response was considered. In [30] it is shown, based on the same data set as considered here, that in a two-step multiple comparison and model selection procedure often also other models than the log-logistic model are chosen, such as the linear model and the non-monotone Beta model. Some of these models directly include the $EC50$ as a parameter; for others, this alert concentration needs to be derived analytically or even numerically. In principle, however, an extension of the approach for information sharing proposed here to other functional relationships is easily possible. Instead of selecting one specific model, model averaging approaches can lead to more accurate estimates [31]. In addition, it is possible to consider other alert concentrations that are not directly included as parameters in the model, such as other ED-values, the BMD or the LEC.

Using the assumption of normal distributed response data for fitting a parametric model via the explained methodology, we implicitly restricted the method for application to appropriately pre-processed microarray data. However, the popular RNA-seq and TempO-Seq technologies lead to counts as outcomes. These are assumed to follow a negative binomial distribution, as seen in the `R`-package `DESeq2` [32] which is used for determining differentially expressed genes. Since fitting concentration-response curves for count data is also possible, e.g. using the `R`-package `drc` [2], an extension of the approach proposed here to other types of data is possible.

One further possible extension of our approach is given by directly incorporating the sharing of information between genes in a hierarchical Bayesian model, thus avoiding the two-step

procedure. However, the approach proposed here benefits from its intuitive interpretability and the easy implementation using standard packages.

## Supporting information

**S1 Fig. Parameter estimates for the VPA dataset.** Estimates of the four parameters $b, c, d, \tilde{e}$ of the 4pLL model, fitted to the 7191 genes selected from the VPA dataset, are shown by histograms.
(PDF)

**S2 Fig. Mixture of 5 normal distributions fitted to the values of parameter $\tilde{e}$.** The corresponding parameter estimates are summarized in Table 2, where the first component corresponds to the red curve, the second component to the green curve, the third component to the blue curve, the fourth component corresponds to the turquoise curve, and the fifth component corresponds to the purple curve.
(PDF)

**S3 Fig. MSE for the direct estimation and the Bayes approach based on robust estimation (A), together with true underlying parameters $b$ and $\tilde{e}$ (B).** The resulting values of the MSE are colored according to the comparative performance of the two approaches. The underlying parameter values are colored in the same way.
(PDF)

**S4 Fig. MSE for the direct estimation and the Bayes approach based on the mixing distribution as prior (A), together with true underlying parameters $b$ and $\tilde{e}$ (B).** The resulting values of the MSE are colored according to the comparative performance of the two approaches. The underlying parameter values are colored in the same way.
(PDF)

**S5 Fig. Parameters from the prior normal distribution, estimated in an empirical way directly from the direct estimates for parameter $\tilde{e}$ in each simulation run separately.** The two histograms on the left (A) show the values for the prior mean of the normal distribution, the two histograms on the right (B) show the values for the prior standard deviation. The parameters are estimated via ML estimation (top) or via robust estimation (bottom).
(PDF)

**S6 Fig. Parameters from the mixed prior normal distribution, estimated in an empirical way directly from the direct estimates for parameter $\tilde{e}$ in each simulation run separately.** The five rows show the individual mixture components, and the columns the mixing parameter $\lambda$ (left), the prior mean (middle) and the prior standard deviation (right) of the respective mixture component.
(PDF)

## Acknowledgments

The authors would like to thank Katja Ickstadt for the helpful discussion about the Bayesian approaches.

## Author Contributions

**Conceptualization:** Franziska Kappenberg, Jörg Rahnenführer.

**Data curation:** Franziska Kappenberg.

**Formal analysis:** Franziska Kappenberg.

**Funding acquisition:** Jörg Rahnenführer.

**Investigation:** Franziska Kappenberg.

**Methodology:** Franziska Kappenberg, Jörg Rahnenführer.

**Resources:** Jörg Rahnenführer.

**Software:** Franziska Kappenberg.

**Supervision:** Jörg Rahnenführer.

**Validation:** Franziska Kappenberg, Jörg Rahnenführer.

**Visualization:** Franziska Kappenberg.

**Writing – original draft:** Franziska Kappenberg.

**Writing – review & editing:** Jörg Rahnenführer.

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
