## [Decision Letter · Decision Letter 0]

23 Aug 2023

PONE-D-23-01563Information sharing in high-dimensional gene expression data for improved parameter estimation in concentration-response modellingPLOS ONE

Dear Dr. Kappenberg,

Thank you for submitting your manuscript to PLOS ONE. After careful consideration, we feel that it has merit but does not fully meet PLOS ONE’s publication criteria as it currently stands. Therefore, we invite you to submit a revised version of the manuscript that addresses the points raised during the review process.

We look forward to receiving your revised manuscript.

Kind regards,

Shamik Polley, Ph.D

Academic Editor

PLOS ONE

Journal Requirements:

Reviewers' comments:

Reviewer's Responses to Questions

**Comments to the Author**

1. Is the manuscript technically sound, and do the data support the conclusions?

Reviewer #1: Yes

Reviewer #2: Yes

Reviewer #3: No

2. Has the statistical analysis been performed appropriately and rigorously? 

Reviewer #1: Yes

Reviewer #2: Yes

Reviewer #3: No

3. Have the authors made all data underlying the findings in their manuscript fully available?

Reviewer #1: Yes

Reviewer #2: Yes

Reviewer #3: No

4. Is the manuscript presented in an intelligible fashion and written in standard English?

Reviewer #1: Yes

Reviewer #2: Yes

Reviewer #3: Yes

5. Review Comments to the Author

Reviewer #1: In the current era, simulation studies are growing in the field of biological studies. Yet, it’s still a growing field. In the same sense, the presented paper proposes a new method to improve the estimation of minimal toxicity. The authors used an approach where information is shared across genes which allows the relaxation of common parameters and hence allows some improvements in the statistical analyses. Empirical Bayes approaches were used to carry out the study. Results were clearly detailed and explained. The overall notion shows significant merit in the application of robust Bayes and mixing distribution Bayes approaches for acquiring more information from certain data. Indeed, this could be useful for several biological models.

Reviewer #2: Your approach to information sharing in high-dimensional gene expression data is both innovative and insightful. The utilization of concentration-response modeling, coupled with the incorporation of information sharing techniques, showcases your expertise in handling complex biological datasets. The results you have obtained demonstrate the potential for improved parameter estimation, which holds promising implications for advancing our understanding of gene expression patterns and their regulatory mechanisms.

I commend your meticulous data analysis and the clarity with which you have presented your findings. Your attention to detail and rigorous methodology inspire confidence in the validity and reliability of your research outcomes. Furthermore, your ability to effectively communicate complex concepts ensures that your work can be understood and appreciated by both fellow researchers and non-experts alike.

In addition to the scientific rigor, your work also exemplifies your passion for the subject matter. It is evident that you possess a genuine curiosity for unraveling the intricacies of gene expression, and this enthusiasm shines through in every aspect of your research.

Reviewer #3: The paper titled "Information sharing in high-dimensional gene expression data for improved parameter estimation in concentration-response modelling" utilizes a four-parameter logistic function to model toxicological concentration-response, a model that was previously proposed. The novelty of this work lies in the application of a Bayesian approach to obtain probabilities (posterior distributions fed by previous information). This approach considers the alert concentration EC50 as a parameter. Ultimately, the results are compared with those obtained using other three different approaches.

Considering that PLOS ONE objectively focuses on the technical aspects of a study rather than subjective evaluations, and after having read the paper I recommend rejecting the current manuscript. My reasons for this recommendation are attached in a file.

6. PLOS authors have the option to publish the peer review history of their article (what does this mean?). If published, this will include your full peer review and any attached files.

Reviewer #1: No

Reviewer #2: **Yes: **Gustavo A Fernandez

Reviewer #3: No

---

## [Author Response · Author response to Decision Letter 0]

8 Sep 2023

Dear Editor, dear Reviewers,

thank you very much for the helpful comments. We have carefully addressed all points in a point-by-point letter, in which all raised comments are discussed in detail. This letter is uploaded alongside the modified manuscript.

Best regards

Franziska Kappenberg and Jörg Rahnenführer

---

## [Decision Letter · Decision Letter 1]

9 Oct 2023

Information sharing in high-dimensional gene expression data for improved parameter estimation in concentration-response modelling

PONE-D-23-01563R1

Dear Dr. Kappenberg,

We’re pleased to inform you that your manuscript has been judged scientifically suitable for publication and will be formally accepted for publication once it meets all outstanding technical requirements.

Kind regards,

Shamik Polley, M.V.Sc (Veterinary Biochemistry); Ph.D (Genetics)

Academic Editor

PLOS ONE

Additional Editor Comments (optional):

Reviewers' comments:

Reviewer's Responses to Questions

**Comments to the Author**

1. If the authors have adequately addressed your comments raised in a previous round of review and you feel that this manuscript is now acceptable for publication, you may indicate that here to bypass the “Comments to the Author” section, enter your conflict of interest statement in the “Confidential to Editor” section, and submit your "Accept" recommendation.

Reviewer #4: All comments have been addressed

2. Is the manuscript technically sound, and do the data support the conclusions?

Reviewer #4: Yes

3. Has the statistical analysis been performed appropriately and rigorously? 

Reviewer #4: Yes

4. Have the authors made all data underlying the findings in their manuscript fully available?

Reviewer #4: Yes

5. Is the manuscript presented in an intelligible fashion and written in standard English?

Reviewer #4: Yes

6. Review Comments to the Author

Reviewer #4: The authors have revised the manuscript properly therefore, the recommendation for publication in its present form is given by my side.

7. PLOS authors have the option to publish the peer review history of their article (what does this mean?). If published, this will include your full peer review and any attached files.

Reviewer #4: No

---

## [Editor Report · Acceptance letter]

12 Oct 2023

PONE-D-23-01563R1 

Information sharing in high-dimensional gene expression data for improved parameter estimation in concentration-response modelling 

Dear Dr. Kappenberg:

I'm pleased to inform you that your manuscript has been deemed suitable for publication in PLOS ONE. Congratulations! Your manuscript is now with our production department. 

Kind regards, 

on behalf of

Dr. Shamik Polley 

Academic Editor

PLOS ONE